# The transcription factor SKN-1 drives lysosomal enlargement during aging to maintain function

**Xinyu Wang[1☯], Huimin Liu[1☯], Xiaoman Wang[1], Ben Zhou[2], Haiqing Tang[1], Shanshan Pang[1]***

**1** School of Life Sciences, Chongqing University, Chongqing, China, **2** Shanghai Institute of Nutrition and Health, University of Chinese Academy of Sciences, Chinese Academy of Sciences, Shanghai, China

☯ These authors contributed equally to this work
* sspang@cqu.edu.cn












## Abstract

Lysosomes are critical hubs for both cellular degradation and signal transduction, yet their function declines with age. Aging is also associated with significant changes in lysosomal morphology, but the physiological significance of these alterations remains poorly understood. Here, we find that a subset of aged lysosomes undergo enlargement resulting from lysosomal dysfunction in *Caenorhabditis elegans (C. elegans)*. Importantly, this enlargement is not merely a passive consequence of functional decline but represents an active adaptive response to preserve lysosomal degradation capacity. Blocking lysosomal enlargement exacerbates the impaired degradation of dysfunctional lysosomes. Mechanistically, lysosomal enlargement is a transcriptionally regulated process governed by the longevity transcription factor SKN-1, which responds to lysosomal dysfunction by restricting fission and thereby induces lysosomal enlargement. Furthermore, in long-lived germline-deficient animals, SKN-1 activation induces lysosomal enlargement, thereby promoting lysosomal degradation and contributing to longevity. These findings unveil a morphological adaptation that safeguards lysosomal homeostasis, with potential relevance for lysosomal aging and life span.

## Introduction

Lysosomes, as centers for degradation and recycling, are crucial for the maintenance of cellular homeostasis. Various intracellular and extracellular factors can impair lysosomal function, which is strongly associated with aging and numerous human diseases [1,2]. Studies in model organisms, such as yeast and *C. elegans*, reveal that lysosomal phenotypes, including acidity and degradation capacity, decline with age [3–5]. Conversely, several long-lived mutants maintain healthy lysosomes and extend life span in a lysosome-dependent manner [5], highlighting the critical role of functional lysosomes in promoting longevity. Therefore, understanding how aged

**Data availability statement:** All relevant data are within the paper and its Supporting Information files.

**Funding:** This work was supported by the National Natural Science Foundation of China (Grant nos. 32271212 and 32071163 [to S.P.] and Grant nos. 32370828 and 32070754 [to H.T.]), the National Key R&D Program of China (Grant no. 2023YFA1801100 to B.Z.), the Fundamental Research Funds for the Central Universities (2024CDJXY016), the Natural Science Foundation of Chongqing, China (Grant no. cstc2021ycjh-bgzxm0138 [to S.P.]). The funders had no role in study design, data collection and analysis, decision to publish, or preparation of the manuscript.

**Competing interests:** The authors have declared that no competing interests exist.

**Abbreviations:** ALs, autolysosomes; APs, autophagosomes; BORC, BLOC-1-related complex; gof, gain-of-function; LLOME, L-leucyl-L-leucine methyl ester; LSDs, lysosomal storage diseases; NAC, N-acetylcysteine; NGM, nematode growth medium; ROS, reactive oxygen species; TBHP, tert-butyl hydroperoxide; TFs, transcription factors; WT, wild-type.

lysosomes preserve their degradation functions is of particular importance, yet this question remains largely unexplored.

Another hallmark of lysosomal aging is extensive morphological alteration [5]. The complex relationship between organellar morphology and function underscores the pivotal role of morphology in dictating organellar function, as exemplified in mitochondrial research [6,7]. Similarly, lysosomes, as highly dynamic organelles, undergo continuous modulation of morphology through fusion and fission events [8,9]. Lysosomes adopt two main morphologies: tubular and vesicular. Aged lysosomes frequently form extensive tubular networks [5], a feature also observed in long-lived diet-restricted animals, where lysosomal tubulation contributes to life span extension [10]. It has been proposed that increased autophagic demand during aging and nutrient deficiency may drive lysosomal tubulation [10]. Vesicular lysosomal enlargement represents another form of morphological change. This enlargement is a hallmark of lysosomal storage diseases (LSDs) [8] and is implicated in the pathogenesis of Parkinson's disease [11]. However, the functional implications of lysosomal size changes during aging remain poorly understood, and its causal relationship with lysosomal function is unclear.

In this study, we investigated the regulation and function of lysosomal morphological changes during aging using *C. elegans* as a model organism. We found that a subset of aged lysosomes undergo enlargement. Unlike lysosomal tubulation, which responds to cellular changes such as increased autophagic demand, lysosomal enlargement is an adaptive response to their own dysfunction, aiming to preserve lysosomal degradation capacity. Remarkably, this adaptive response is governed by transcriptional regulation through SKN-1, known for its role in promoting longevity. Consistent with these findings, SKN-1-mediated lysosomal enlargement improves lysosomal function and extends life span in germline-deficient, long-lived *C. elegans*, suggesting a potential connection between lysosomal enlargement and aging.

## Results

### Aging is associated with lysosomal enlargement

Aging is associated with significant changes in lysosomal morphology in *C. elegans*. Aged lysosomes often form extensive tubular networks [5]. Using the lysosomal fluorescent reporter NUC-1::mCHERRY [12] in the hypodermis, we also observed that lysosomes in aged *C. elegans* frequently adopted a heavily tubular morphology; however, this alteration was evident only in a subset of animals (Fig 1A and 1B), while lysosomes in most of the remaining animals retained a vesicular morphology (Fig 1B and 1C). We observed three distinct types of lysosomal morphology in aged animals: heavily tubular, vesicular, and a mixed tubular-vesicular form. Animals with heavily tubular lysosomes were most prevalent in the middle-aged stage (day 5 adults) but declined in the late-aged stage (day 9 adults) (Fig 1B). In contrast, the proportion of animals exhibiting a vesicular morphology increased with age, with more observed on day 9 compared to day 5 (Fig 1B). More importantly, these vesicular lysosomes in aged animals were significantly enlarged compared to those in young adults (day 1) (Fig 1C and 1D), suggesting a strong correlation between aging and lysosomal enlargement.

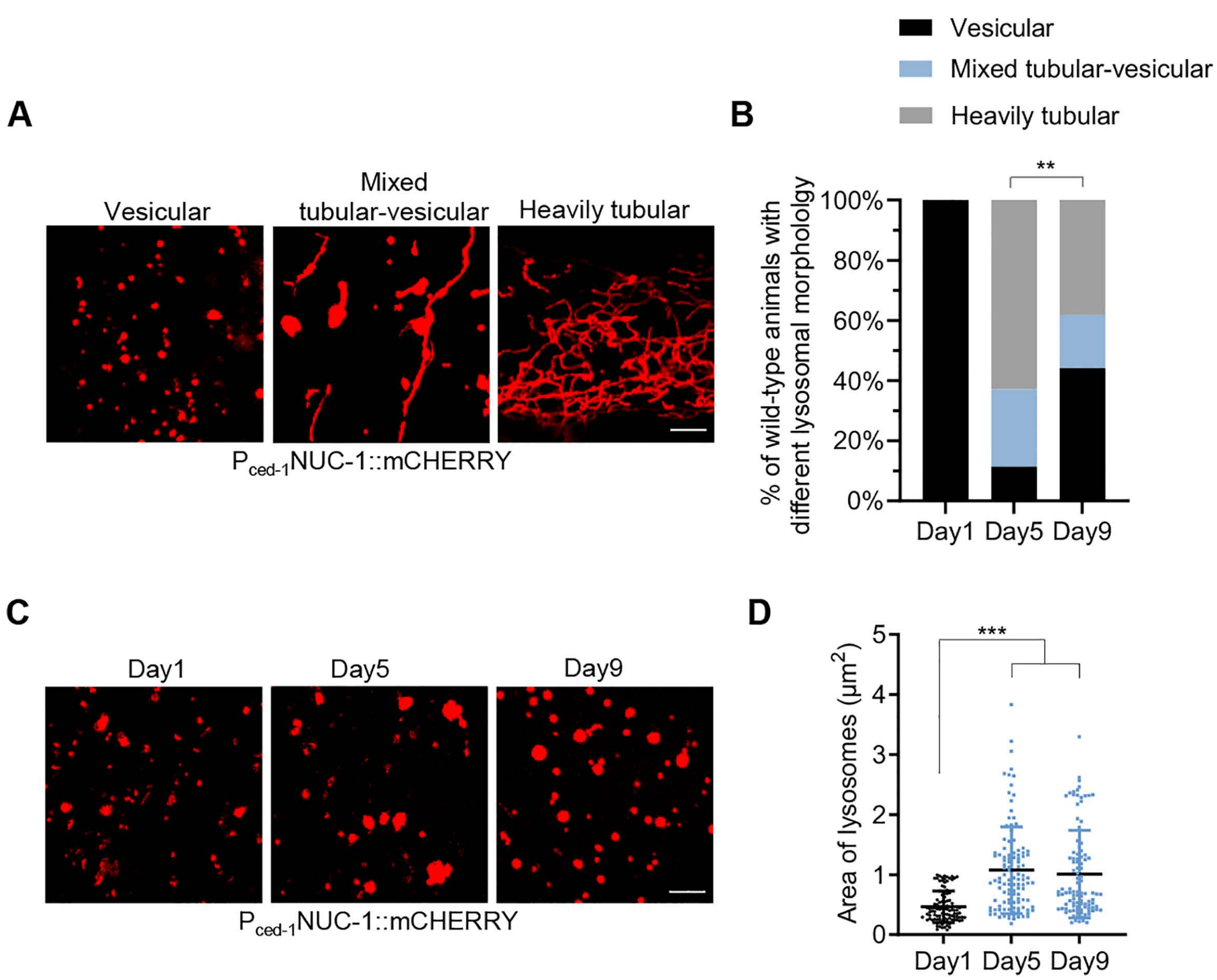


**Fig 1. Aging is associated with lysosomal enlargement. (A)** Representative images of three different lysosomal morphologies in the hypodermis during aging. **(B)** Ratio of animals exhibiting different lysosomal morphologies in the hypodermis during aging. Chi-square and Fisher's exact test. $n = 34–35$ animals. **(C** and **D)** The morphology (C) and size (D) of vesicular lysosomes in the hypodermis during aging. One-way ANOVA analysis followed by Dunnett's multiple comparisons post hoc test. $n = 102–112$ lysosomes. Data are presented as mean ± SD. $^{**}p < 0.01$, $^{***}p < 0.001$. Scale bar = 5 μm for panels (A) and (C). The numerical data presented in this figure can be found in S1 Data.


To further substantiate this conclusion, we analyzed lysosomal morphology in the intestine and muscle. Using the SPIN-1::mCHERRY reporter for intestinal lysosomes [10], we observed an increase in the proportion of vesicular lysosomes at day 9 compared to day 5 (S1A Fig), accompanied by concurrent enlargement of vesicular lysosomes (S1B Fig). These were consistent with hypodermal findings. Additionally, muscle lysosomes visualized via LAAT-1::GFP [13] exhibited extremely faint fluorescence in tubular structures, precluding reliable semi-quantitative analysis. Nevertheless, vesicular lysosomes in muscle also enlarged during aging (S1C Fig). Collectively, these results confirm that aging is associated with lysosomal enlargement across multiple tissues.

## Lysosomal dysfunction is associated with lysosomal enlargement during aging

Numerous studies have shown that lysosomal enlargement is often associated with their functional decline, as seen in LSDs [8]. This led us to speculate that lysosomal enlargement may occur as an active response to lysosomal intrinsic changes during aging, while tubulation may represent a response to cellular demands. We further proposed that if lysosomal enlargement is indeed an active response to lysosomal dysfunction, rather than a passive consequence of

functional decline, key evidence in support of this hypothesis would include: (1) lysosomal dysfunction does lead to their enlargement; (2) this process is governed by a regulatory mechanism; and (3) lysosomal enlargement is beneficial for their degradation capacity.

We thus explored whether lysosomal dysfunction leads to their enlargement during aging. Aging is associated with a decline in lysosomal function in *C. elegans* [5], which we confirmed by assessing the activity of acid phosphate (Fig 2A), a predominantly lysosomal enzyme, while acknowledging the potential contribution from nonlysosomal sources. We then examined the involvement of reactive oxygen species (ROS) in this process, as they play a crucial role of in aging [14] and are also able to induce lysosomal dysfunction [15]. Through treatment with the ROS scavenger N-acetylcysteine (NAC), we found that acid phosphate activity was partially restored during aging (Fig 2A), implicating ROS as a contributor to age-related lysosomal dysfunction. Moreover, NAC also partially prevented lysosomal enlargement in aged animals (Fig 2B and 2C), suggesting that lysosomal impairment by ROS may contribute to lysosomal enlargement during aging. It should also be noted that the acid phosphatase assay measured the enzyme activity from the whole animal, reflecting the combined activity of both vesicular and tubular lysosomes.

Next, we asked if ROS is sufficient to cause lysosomal enlargement in young animals. Tert-butyl hydroperoxide (TBHP) is a commonly used ROS inducer in *C. elegans*. We found that TBHP treatment impaired lysosomal acid phosphate activity, progressively worsening after three hours of treatment (S2A Fig). This ROS-induced lysosomal dysfunction after 12 hours was confirmed by NUC-1::mCHERRY cleavage (Fig 2D), where the quantification of cleaved mCHERRY, processed by lysosomal cathepsins, indicates lysosomal function. Together, these data suggest that ROS is sufficient to cause lysosomal dysfunction. Next, using the lysosomal fluorescent reporter NUC-1::mCHERRY, we found that lysosomes subjected to TBHP treatment exhibited gradual enlargement in the hypodermis (Fig 2E and 2F), corresponding to their functional decline. Similar effects were observed upon treatment with hydrogen peroxide, another ROS (S2B and S2C Fig). Thus, ROS exposure indeed causes lysosomal enlargement.

NUC-1::mCHERRY expression is driven by the engulfing cell-specific *ced-1* promoter. While NUC-1::mCHERRY expression is predominantly localized to the hypodermis, it is also detected in a limited number of other cells [12]. We therefore extended our analysis to the tissue level to correlate ROS-induced lysosomal dysfunction with enlargement. Given that lysosomes degrade autophagic cargo after fusion with autophagosomes (APs), we employed the mCHERRY::GFP::LGG-1 strain [16] to monitor autophagic cargo turnover as an indicator of lysosomal degradative capacity. LGG-1 is an AP protein and GFP fluorescence is quenched in acidic environment after AP fusion with lysosomes, thus a decrease of the GFP/mCHERRY ratio indicates enhanced autophagic flux efficiency. Our results revealed that ROS treatment elevated the GFP/mCHERRY ratio in the hypodermis (S2D Fig), intestine (S2E Fig), and muscle (S2F Fig), indicative of compromised lysosomal degradation. Correspondingly, ROS exposure induced lysosomal enlargement not only in the hypodermis (Fig 2E and 2F) but also in the intestine (S2G and S2H Fig) and muscle (S2I and S2J Fig). These observations demonstrate that ROS-induced lysosomal dysfunction is associated with morphological enlargement across diverse tissues.

Next, we examined the lysosomal proton pump vacuolar $H^+$-ATPase (V-ATPase), as lysosomal function depends on its luminal acidic environment and the acidity of lysosomes decreases during *C. elegans* aging [4,5]. By using RNAi targeting multiple subunits of *C. elegans* V-ATPase, we found that inhibition of V-ATPase also caused lysosomal enlargement in the hypodermis (Fig 2G and 2H), further suggesting that aging-related lysosomal impairment can lead to their enlargement. Additionally, we were curious whether such lysosomal enlargement is specific to aging-related stresses or represents a more generalized response to other forms of lysosomal dysfunction. To test this, we treated animals with L-leucyl-L-leucine methyl ester (LLOME), a reagent known to cause lysosomal membrane damage, and observed both lysosomal dysfunction (S2K Fig) and enlargement (Fig 2I and 2J). Together, these data suggest that aging-related lysosomal impairment causes lysosomal enlargement, which might be a general response to multiple forms of lysosomal dysfunction.

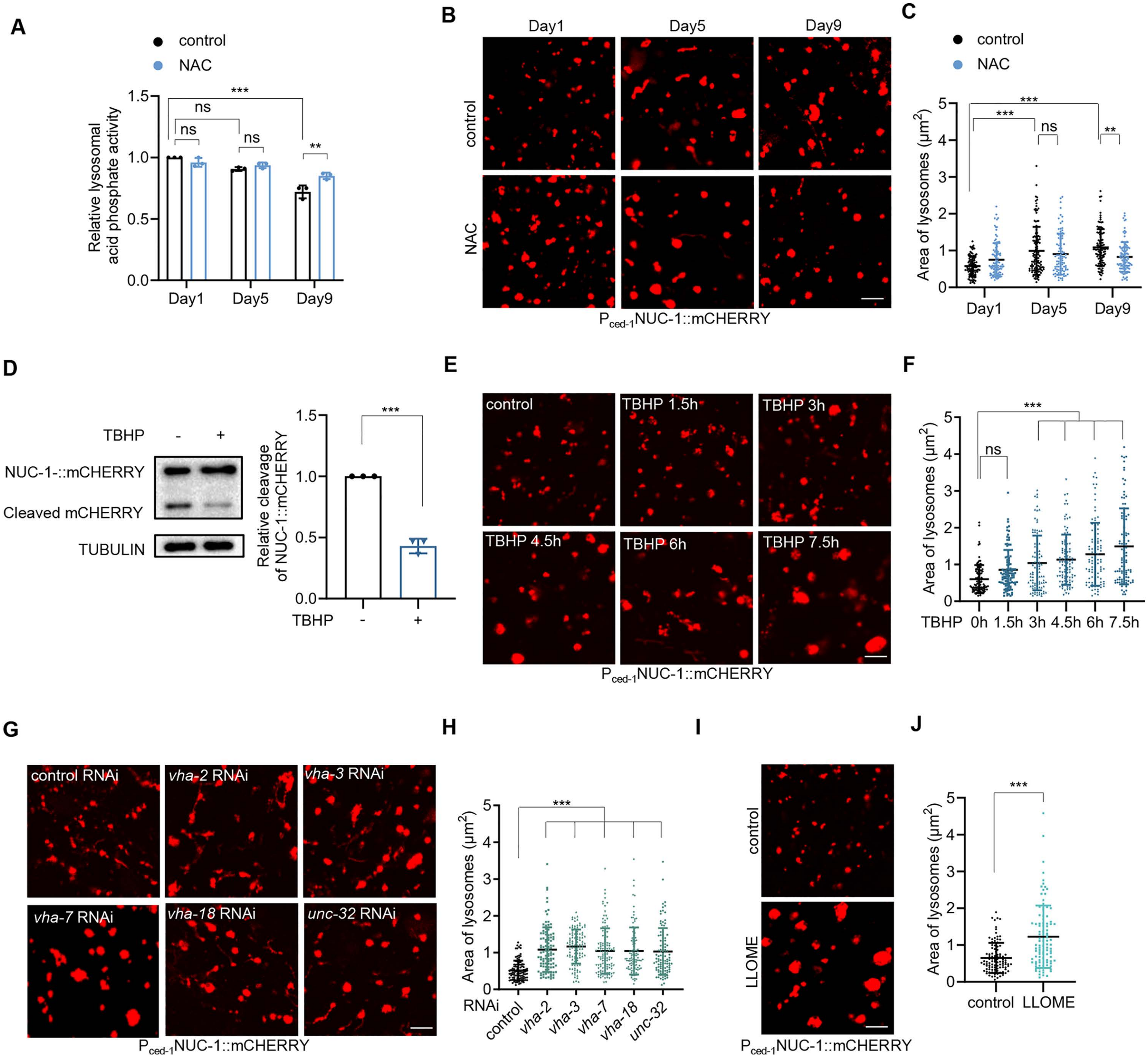


**Fig 2. Lysosomal morphological changes in response to lysosomal dysfunction. (A)** Effect of NAC treatment on the activity of lysosomal acid phosphate during aging. Two-way ANOVA analysis followed by Tukey's multiple comparisons post hoc test. $n$ = three independent experiments. **(B and C)** Effect of NAC treatment on the lysosomal morphology (B) and size (C) in the hypodermis during aging. Two-way ANOVA analysis followed by Tukey's multiple comparisons post hoc test. $n$ = 84–114 lysosomes. **(D)** Effect of 12-hour TBHP treatment on NUC-1::mCHERRY cleavage in day 1 adults. Left: representative images. Right: quantification data. Unpaired $t$ test analysis. $n$ = three independent experiments. **(E and F)** Effect of TBHP treatment on lysosomal morphology (E) and size (F) in the hypodermis of day 1 adults. One-way ANOVA analysis followed by Dunnett's multiple comparisons post hoc test. $n$ = 100–101 lysosomes. **(G and H)** Effects of RNAi targeting V-ATPase subunits on lysosomal morphology (G) and size (H) in the hypodermis of day 1 adults. One-way ANOVA analysis followed by Dunnett's multiple comparisons post hoc test. $n$ = 98–114 lysosomes. **(I and J)** Effect of 6-hour treatment with LLOME on lysosomal morphology (I) and size (J) in the hypodermis of day 1 adults. Unpaired $t$ test analysis. $n$ = 100 lysosomes. Data are presented as mean ± SD. ** $p < 0.01$, *** $p < 0.001$. Scale bar = 5 μm for panels (B), (E), (G), and (I). The numerical data presented in this figure can be found in S1 Data. Immunoblot raw images in this figure can be found in S1 File.


### The transcription factor SKN-1 is a critical regulator of lysosomal enlargement

Next, we explored whether lysosomal enlargement is governed by a regulatory mechanism. We utilized the ROS-induced lysosomal enlargement model and hypothesized the involvement of a transcriptional program, as lysosomal enlargement occurs hours after ROS exposure. Thus, we conducted three rounds of RNAi screening targeting 757 predicted transcription factors (TFs) in *C. elegans* to identify RNAi clones that suppress lysosomal enlargement induced by ROS treatment (Fig 3A). This approach revealed SKN-1 as the sole TF whose knockdown robustly and consistently inhibited lysosomal enlargement in TBHP-exposed animals. As the *C. elegans* ortholog of mammalian Nrf, SKN-1 is known for its roles in oxidative stress response, lipid metabolism, and longevity [17]. *skn-1* RNAi substantially suppressed the enlargement of lysosomes in response to ROS (Fig 3B and 3C). This effect was confirmed by using *skn-1* mutants (S3A and S3B Fig). In contrast, RNAi targeting *hlh-30*, a central TF governing lysosomal biogenesis [18], had no effects on ROS-induced lysosomal enlargement (S3C and S3D Fig). We further examined the role of SKN-1 in lysosomal enlargement during aging and found that *skn-1* RNAi or mutants substantially reduced vesicular lysosomal enlargement in aged animals (Figs 3D, 3E, S3E, and S3F). Additionally, *skn-1* RNAi did not affect the proportions of vesicular and tubular lysosomes in aged animals (S3G Fig). These data collectively suggest that SKN-1 is essential for lysosomal enlargement in response to lysosomal dysfunction and aging.

We next asked whether SKN-1 is activated in response to lysosomal dysfunction. ROS-induced the expression of the SKN-1 transcriptional reporter *gst-4p*::GFP (S3H Fig) as reported [19]. Our results demonstrated that RNAi targeting V-ATPase subunits (Fig 3F) or LLOME treatment (S3I Fig) also activated the SKN-1 reporter *gst-4p*::GFP, suggesting that SKN-1 responds broadly to lysosomal dysfunction.

To further support the critical role of SKN-1 in governing lysosomal enlargement, we tested if SKN-1 activation is sufficient to promote lysosomal enlargement in the absence of lysosomal dysfunction. We employed *skn-1(lax120)* animals, a *skn-1* gain-of-function (gof) mutant [20], and observed the enlargement of vesicular lysosomes (Fig 3G and 3H), whereas the proportions of vesicular and tubular lysosomes during aging remained unaltered (S3J Fig). Additionally, RNAi targeting *wdr-23*, a negative regulator of *skn-1* [21], also induced similar effects (S3K and S3L Fig). These findings collectively suggest that SKN-1 is the critical TF that governs lysosomal enlargement in response to their dysfunction.

### Lysosomal fission is restricted in response to lysosomal dysfunction

We next sought to understand the underlying reasons for lysosomal enlargement. Lysosomal fission allows the regeneration of smaller lysosomes from larger ones, thereby reducing lysosomal size (Fig 4A). Indeed, inhibition of fission in *C. elegans* results in the accumulation of enlarged lysosomes [22,23]. We thus proposed that lysosomal dysfunction might limit fission, resulting in lysosomal enlargement.

Lysosomal fission is driven by the kinase PIKfyve via the production of phosphatidylinositol-3,5-bisphosphate [PI(3,5)P2] in the lysosomal membrane [24]. In *C. elegans*, PPK-3, the ortholog of PIKfyve, serves as the primary regulator of lysosomal fission [22]. We found that RNAi targeting *ppk-3* caused the accumulation of enlarged lysosomes (Fig 4B and 4C). Conversely, the BLOC-1-related complex (BORC) is a known negative regulator of lysosomal fission. Its deletion reduces lysosomal size by inducing fission in mammalian cells [25]. We hypothesized that if fission limitation underlies lysosomal enlargement, then BORC inhibition might reverse this effect. The core subunit of BORC, myrlysin, associates with the lysosomal membrane via its myristoyl group [26]. SAM-4 is the worm ortholog of myrlysin. We found that *sam-4* RNAi reversed lysosomal enlargement in ROS-treated animals (Fig 4D and 4E), aged animals (Fig 4F and 4G), and *skn-1 gof* animals (S4A and S4B Fig). Additionally, it did not affect the proportions of vesicular and tubular lysosomes in aged animals (S4C Fig). Notably, using tissue-specific RNAi strains, we found that *sam-4* RNAi specifically in the hypodermis suppressed ROS-induced hypodermal lysosomal enlargement (S4D and S4E Fig), while RNAi in the intestine and muscle did not (S4F–S4I Fig), suggesting that SAM-4 regulates lysosomal enlargement in a cell-autonomous manner. Together, these results suggest that lysosomal enlargement in lysosome-impaired or aged animals is likely a consequence of fission limitation.

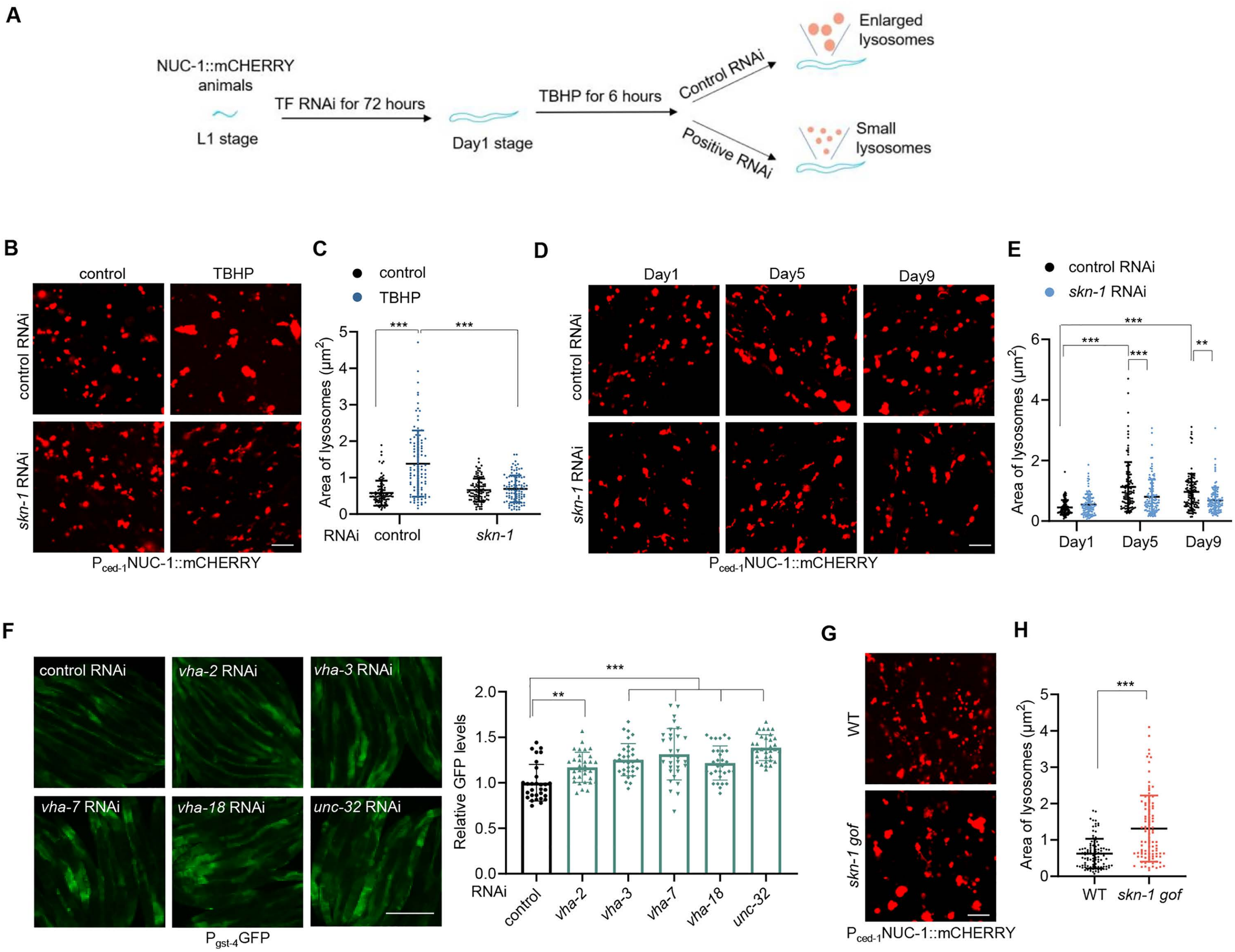


**Fig 3. SKN-1 activation mediates lysosomal enlargement. (A)** Procedure for screening TFs that govern lysosomal enlargement. **(B and C)** Effect of *skn-1* RNAi on lysosomal morphology (B) and size (C) in response to 6-hour TBHP treatment in the hypodermis of day 1 adults. Two-way ANOVA analysis followed by Tukey's multiple comparisons post hoc test. $n = 100$ lysosomes. **(D and E)** Effect of *skn-1* RNAi on vesicular lysosomal morphology (D) and size (E) in the hypodermis during aging. Two-way ANOVA analysis followed by Tukey's multiple comparisons post hoc test. $n = 106-110$ lysosomes. **(F)** RNAi targeting V-ATPase subunits induces expression of *gst-4p*::GFP in day 1 adults. Left: representative images. Right: quantification of fluorescent intensity. One-way ANOVA analysis followed by Dunnett's multiple comparisons post hoc test. $n = 30-32$ animals. **(G and H)** Effect of *skn-1 gof* mutation on lysosomal morphology (G) and size (H) in the hypodermis of day 1 adults. Unpaired *t* test analysis. $n = 100$ lysosomes. Data are presented as mean ± SD. $^{**}p < 0.01$, $^{***}p < 0.001$. Scale bar = 5 μm for panels (B), (D), and (G); 150 μm for panel (F). The numerical data presented in this figure can be found in S1 Data.


The effect of BORC on lysosomes depends on its association with lysosomal membrane, which is mediated by SAM-4/myrlysin [26–28]. We thus asked whether the association between SAM-4 and lysosomes changes under conditions of lysosomal enlargement. To explore this, we generated *sam-4::gfp* transgenic animals driven by the *sam-4* promoter. However, GFP fluorescent signals were undetectable in the hypodermis, where the lysosomal reporter NUC-1::mCHERRY is expressed, likely due to low expression levels. To address this, we created another *sam-4::gfp* transgenic strain using the

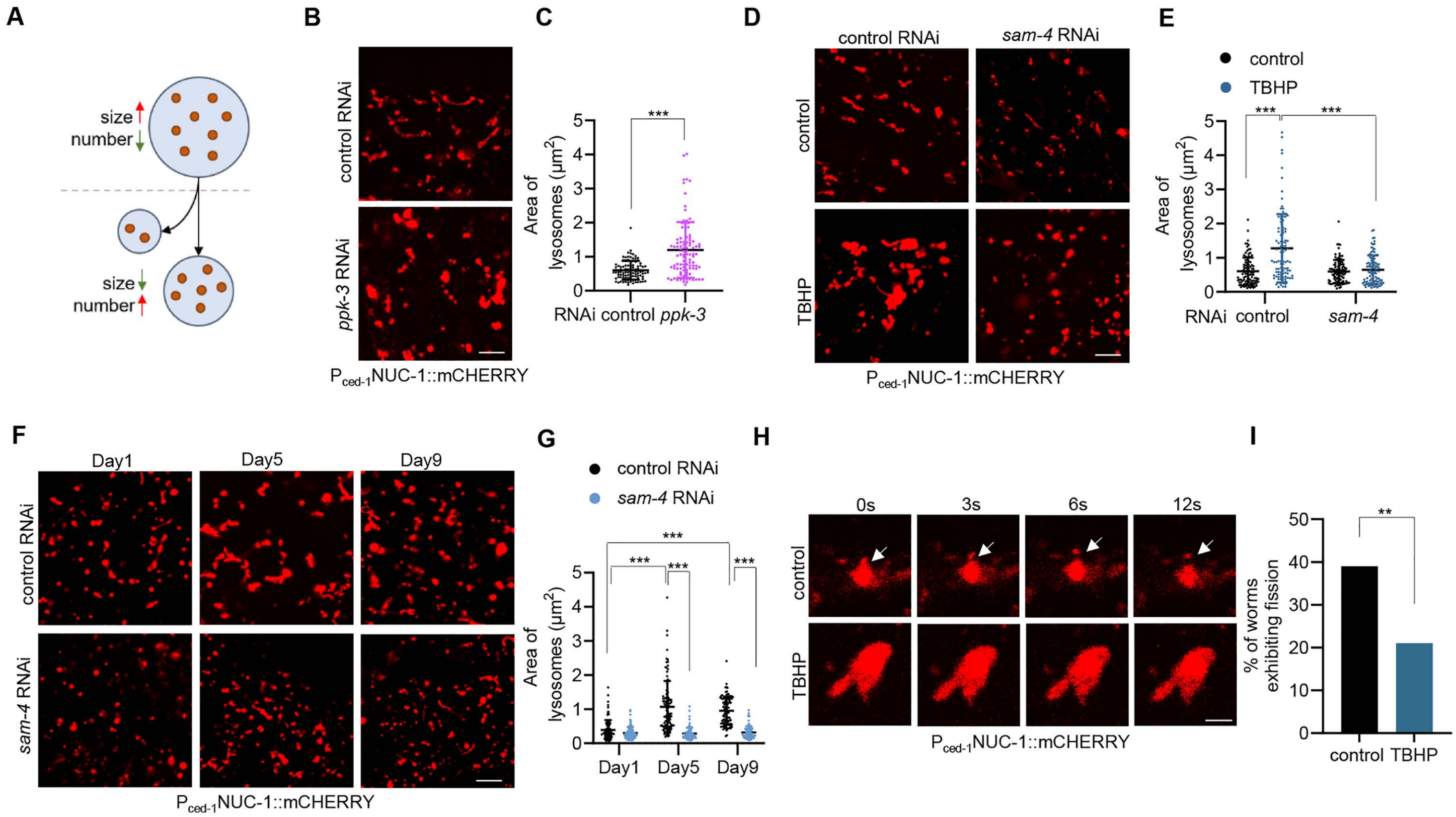


**Fig 4. Lysosomal fission is limited in response to lysosomal dysfunction. (A)** Schematic of lysosomal fission and its impact on lysosomal size and number. **(B** and **C)** Effect of *ppk-3* RNAi on lysosomal morphology (B) and size (C) in the hypodermis of day 1 adults. Unpaired *t* test analysis. $n = 98$ lysosomes. **(D** and **E)** Effect of RNAi targeting *sam-4* on lysosomal morphology (D) and size (E) in the hypodermis of day 1 adults in response to 6-hour TBHP treatment. Two-way ANOVA analysis followed by Tukey's multiple comparisons post hoc test. $n = 100$ lysosomes. **(F** and **G)** Effect of RNAi targeting *sam-4* on vesicular lysosomal morphology (F) and size (G) in the hypodermis during aging. Two-way ANOVA analysis followed by Tukey's multiple comparisons post hoc test. $n = 100$ lysosomes. **(H)** Representative images of hypodermal lysosomal fission in WT and 6-hour TBHP-treated day 1 adults. White arrows indicate lysosome undergoing fission. **(I)** Quantification of lysosomal fission in response to 6-hour TBHP treatment in the hypodermis of day 1 adults. Chi-square and Fisher's exact test. $n = 20–26$ animals. Data are presented as mean ± SD. $**p < 0.01$, $***p < 0.001$. Scale bar = 5 μm for panels (B), (D), and (F); 1.25 μm for panel (H). The numerical data presented in this figure can be found in S1 Data.


strong hypodermal promoter *dpy-7p* and successfully observed GFP signals in the hypodermis. In WT animals, SAM-4::GFP signals were rarely detected around lysosomes (S4J and S4K Fig). However, upon ROS exposure (S4J Fig) or *skn-1 gof* (S4K Fig), SAM-4 puncta were frequently associated with lysosomes. Thus, while further biochemical analysis is needed to determine whether SAM-4 directly interacts with lysosomal membranes, these data suggest that enlarged lysosomes are associated with the BORC subunit SAM-4.

We then directly monitored the fission events using ROS-treated animals and *skn-1 gof* mutants as representatives. Results revealed that while fission was often observed in control animals, it was rarely seen in the enlarged lysosomes of animals with ROS treatment (Fig 4H), leading to a significant reduction in overall fission events (Fig 4I). Similar fission limitation was also observed in *skn-1 gof* animals (S4L Fig). Additionally, restriction of lysosomal fission may lead to a concomitant reduction in the number of lysosomes (Fig 4A). In line with this, ROS treatment reduced lysosome numbers, which were reversed by *skn-1* RNAi (S4M Fig) and *sam-4* RNAi (S4N Fig). Similarly, *skn-1 gof* mutants exhibited decreased lysosome numbers (S4O Fig).

### Lysosomal enlargement maintains lysosomal degradation capacity

We next explored whether lysosomal enlargement is beneficial for its function. Lysosomes primarily function in the degradation of cellular waste received from APs. During this process, regular lysosomes fuse with APs to form secondary lysosomes, termed autolysosomes (ALs). As ALs are the sites where lysosomal degradation occurs, we first asked if the ALs also enlarged during lysosomal dysfunction. To examine ALs, we assessed the mCHERRY::GFP::LGG-1 reporter. Since LGG-1 is an AP marker protein, this reporter distinguishes ALs with mCHERRY and APs with both GFP and mCHERRY, as GFP fluorescence is quenched in acidic ALs [16]. Our results showed that mCHERRY-positive ALs were enlarged upon ROS exposure in various tissues, which was reversed by *skn-1* RNAi and *sam-4* RNAi (Figs 5A, S5A, and S5B Fig). Aging also increased AL size across various tissues in a SKN-1-dependent manner (Figs 5B, S5C, and S5D Fig). Additionally, *skn-1* activation and *ppk-3* knockdown animals exhibited enlarged ALs in these tissues in the absence of lysosomal dysfunction (Figs 5C, 5D, and S5E–S5H). These results suggest that in response to lysosomal dysfunction, ALs enlarge like regular lysosomes.

We then examined lysosomal degradation using the cleavage of GFP::LGG-1 [29], where AP-derived GFP is cleaved within ALs, serving as an indicator of autophagic cargo degradation by lysosomes. We reasoned that if enlargement facilitates lysosomal degradation, then inhibition of *skn-1* or *sam-4*, which suppress enlargement, should impair lysosomal degradation. As expected, ROS treatment impaired GFP::LGG-1 cleavage (Fig 5E and 5F). Notably, the cleavage of GFP::LGG-1 was further compromised upon *skn-1* or *sam-4* knockdown (Fig 5E and 5F). Conversely, *skn-1 gof* animals exhibited enhanced GFP::LGG-1 cleavage even in the absence of lysosomal dysfunction (Fig 5G), which was suppressed by *sam-4* RNAi (Fig 5G). Additionally, *ppk-3* RNAi, which enlarged lysosomes (Fig 4B and 4C), also enhanced the degradation of GFP::LGG-1 by lysosomes (Fig 5H), suggesting that fission restriction can improve lysosomal degradation. Altogether, these data suggest that lysosomal enlargement facilitates lysosomal degradation, likely depending on fission limitation.

Given that GFP::LGG-1 cleavage indicates lysosomal degradation capacity at the organismal level, and it cannot discriminate between the specific contributions of vesicular versus tubular lysosomes during aging, we further assessed the functional consequence of vesicular lysosomal enlargement at a tissue level. To do this, we analyzed autophagic cargo turnover using the mCHERRY::GFP::LGG-1 strain. A decrease of the GFP/mCHERRY ratio indicates enhanced autophagic flux efficiency. Our results showed that ROS treatment elevated the GFP/mCHERRY ratio across the hypodermis (Fig 5I), intestine (S6A Fig), and muscle tissues (S6B Fig), indicating impaired lysosomal degradation. Critically, these effects were further enhanced by *skn-1* RNAi and *sam-4* RNAi (Figs 5I, S6A, and S6B). Aging also increased the GFP/mCHERRY ratio in various tissues, which was further increased by *skn-1* RNAi (Figs 5J, S6C, and S6D). Conversely, both *skn-1* activation (Figs 5K, S6E, and S6F) and *ppk-3* knockdown (Figs 5L, S6G, and S6H), which promote lysosomal enlargement, reduced the GFP/mCHERRY ratio. These cellular-level observations corroborate conclusions derived from GFP::LGG-1 cleavage assays, confirming that lysosomal enlargement enhances degradative capacity.

### Lysosomal enlargement enhances degradation and longevity in germline-deficient animals

Next, we examined lysosomal enlargement in long-lived mutants and its potential role in longevity regulation. Lysosomal morphology has been well characterized in long-lived *daf-2* (insulin/IGF-1 receptor deficient), *eat-2* (dietary restriction model), and *isp-1* (inhibition of mitochondrial respiration) mutants [5]. These mutants showed varying degrees of lysosomal tubulation, but their vesicular lysosomal size does not increase with age [5]. Another important longevity model in *C. elegans* is the germline-deficient *glp-1* mutant [30], which promotes somatic maintenance and extends life span via SKN-1 activation [31]. Surprisingly, we observed that *glp-1* mutants predominantly displayed highly vesicular lysosomal morphology, with only a few animals showing tubular lysosomes during aging (Fig 6A). Moreover, their vesicular lysosomes were already enlarged at a young age compared to wild-type (WT) controls (Fig 6B and 6C). SKN-1 is also activated in *daf-2*

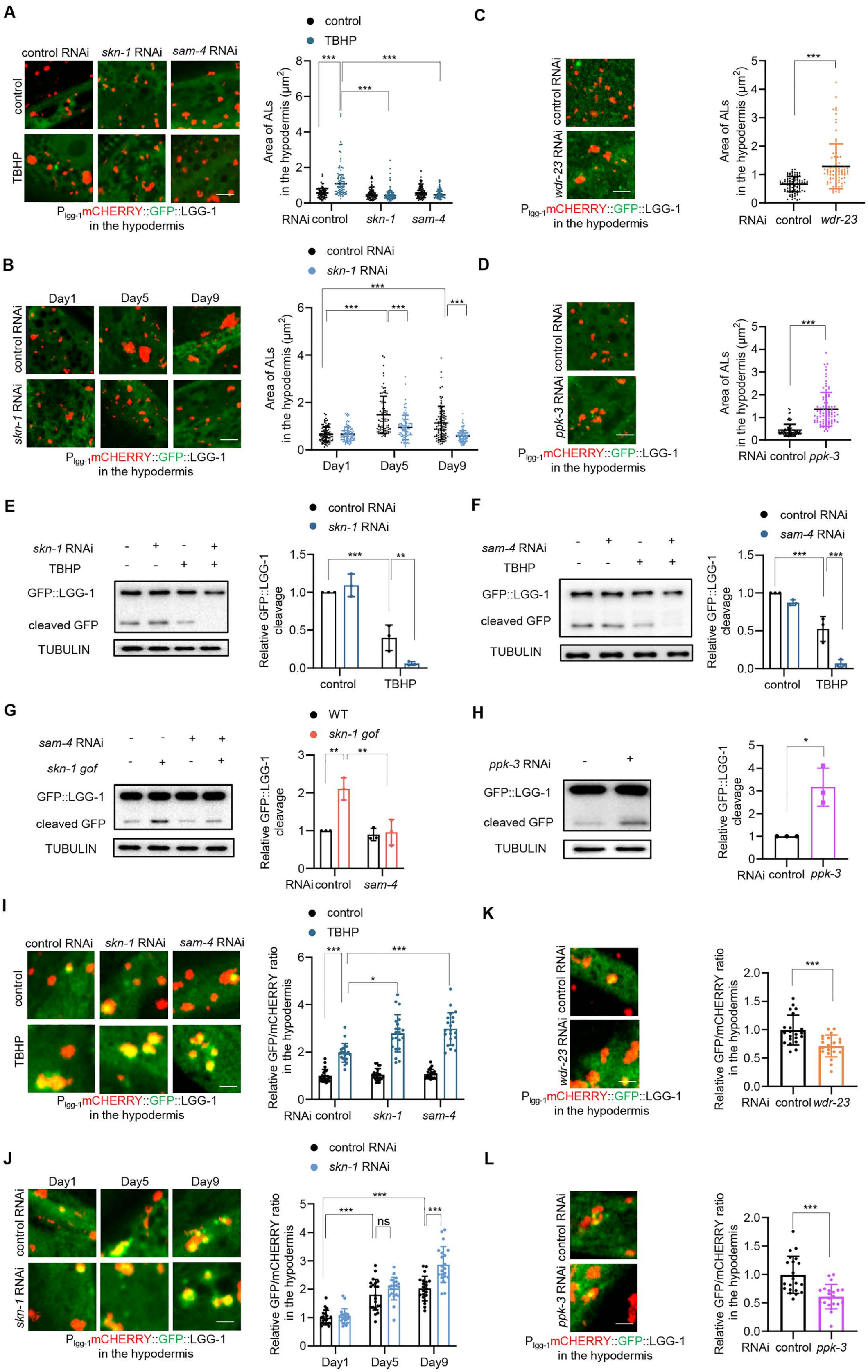
A
control RNAi
skn-1 RNAi
sam-4 RNAi
control
TBHP
Plgg-1mCHERRY::GFP::LGG-1
in the hypodermis
control
TBHP
Area of ALs in the hypodermis (μm²)
RNAi control
skn-1
sam-4
B
Day1
Day5
Day9
control RNAi
skn-1 RNAi
C
wdr-23 RNAi
D
ppk-3 RNAi
E
GFP::LGG-1
cleaved GFP
TUBULIN
Relative GFP::LGG-1 cleavage
F
G
WT
skn-1 gof
H
I
Relative GFP/mCHERRY ratio in the hypodermis
J
ns
K
L

**Fig 5. Lysosomal enlargement preserves lysosomal degradation capacity. (A)** Effects of *skn-1* and *sam-4* RNAi on AL size in the hypodermis of day 1 adults in response to 6-hour TBHP treatment. Left: representative images. Right: quantification data. Two-way ANOVA analysis followed by Tukey's multiple comparisons post hoc test. $n = 90-105$ lysosomes. **(B)** Effect of *skn-1* RNAi on AL size in the hypodermis adults during aging. Left: representative images. Right: quantification data. Two-way ANOVA analysis followed by Tukey's multiple comparisons post hoc test. $n = 89-94$ lysosomes. **(C** and **D)** Effect of *wdr-23* RNAi (C) and *ppk-3* RNAi (D) on AL size in the hypodermis of day 1 adults. Left: representative images. Right: quantification data. Unpaired *t* test analysis. $n = 89-94$ lysosomes for (C) and 91−92 lysosomes for (D). **(E** and **F)** Effect of *skn-1* (E) or *sam-4* (F) RNAi on GFP::LGG-1 cleavage in response to 24-hour TBHP treatment in day 1 adults. Left: representative images. Right: quantification of GFP::LGG-1 cleavage. Two-way ANOVA analysis followed by Tukey's multiple comparisons post hoc test. *n* = three independent experiments. **(G)** Effect of *sam-4* RNAi on GFP::LGG-1 cleavage in response to *skn-1 gof* mutation in day 1 adults. Left: representative images. Right: quantification of GFP::LGG-1 cleavage. Two-way ANOVA analysis followed by Tukey's multiple comparisons post hoc test. *n* = three independent experiments. **(H)** Effect of *ppk-3* RNAi on GFP::LGG-1 cleavage in day 1 adults. Left: representative images. Right: quantification of GFP::LGG-1 cleavage. Unpaired *t* test analysis. *n* = three independent experiments. **(I)** Effects of *skn-1* and *sam-4* RNAi on the GFP/mCHERRY ratio of hypodermal mCHERRY::GFP::LGG-1 in day 1 adults in response to 6-hour TBHP treatment. Left: representative images. Right: quantification data. Two-way ANOVA analysis followed by Tukey's multiple comparisons post hoc test. $n = 21$ animals. **(J)** Effect of *skn-1* RNAi on the GFP/mCHERRY ratio of hypodermal mCHERRY::GFP::LGG-1 during aging. Left: representative images. Right: quantification data. Two-way ANOVA analysis followed by Tukey's multiple comparisons post hoc test. $n = 21$ animals. **(K** and **L)** Effect of *wdr-23* RNAi (K) and *ppk-3* RNAi (L) on the GFP/mCHERRY ratio of hypodermal mCHERRY::GFP::LGG-1 in day 1 adults. Left: representative images. Right: quantification data. Unpaired *t* test analysis. $n = 21$ animals. Data are presented as mean ± SD. *$p < 0.05$, **$p < 0.01$, ***$p < 0.001$. Scale bar = 2.5 μm for panels (A–D); 1.25 μm for panels (I–L). The numerical data presented in this figure can be found in S1 Data. Immunoblot raw images in this figure can be found in S1 File.


mutants [32], but their lysosomal sizes remained comparable to WT controls (S7A and S7B Fig). Thus, lysosomal enlargement appears to be a unique feature of *glp-1* mutants.

Lysosomal function plays a critical role in determining longevity. As such, we hypothesized that SKN-1 activation in *glp-1* mutants promotes lysosomal enlargement, enhances lysosomal degradation, and contributes to longevity. Consistent with this, *skn-1* or *sam-4* RNAi suppressed the lysosomal enlargement of *glp-1* mutants (Fig 6D–6G), suggesting that SKN-1 may induce lysosomal enlargement via fission limitation in these animals. Correspondingly, the enhanced lysosomal degradation in *glp-1* animals, as measured by GFP::LGG-1 cleavage, was inhibited by *skn-1* RNAi (Fig 6H) or *sam-4* RNAi (Fig 6I). These findings suggest that SKN-1-governed lysosomal enlargement enhances lysosomal function in long-lived *glp-1* animals.

While SKN-1 is essential for the longevity of *glp-1* mutants, its effects extend beyond lysosomes [17]. To determine whether lysosomal enlargement contributes to longevity, we examined the effects of *sam-4* RNAi. Our results showed that *sam-4* RNAi shortened the life span of both WT and *glp-1* animals (Figs 6J and 6K), but did not affect the life span of *daf-2* mutants (Fig 6L). These results suggest that SAM-4 is not a general regulator of life span but is specifically required for the longevity of WT and *glp-1* mutants. Considering that only aged WT and *glp-1* mutants exhibit lysosomal enlargement, whereas *daf-2* mutants do not, these findings suggest that lysosomal enlargement may contribute to longevity. Additionally, we evaluated the effect of tissue-specific *sam-4* RNAi on life span in WT *C. elegans*. Knockdown of *sam-4* in the hypodermis, intestine, or muscle tissues each modestly reduced life span (S7C–S7E Fig). This observation aligns with the lysosomal enlargement in all three tissues during aging, suggesting that they all contribute to organismal life span regulation.

## Discussion

In this study, we demonstrate that lysosomal enlargement during aging represents an adaptive response to functional decline. Lysosomal dysfunction activates the transcription factor SKN-1, which promotes lysosomal enlargement by limiting fission. This enlargement helps preserve lysosomal degradation capacity. In long-lived, germline-deficient mutants, SKN-1 is activated even in the absence of lysosomal impairment, enhancing lysosomal degradation and contributing to life span extension (Fig 6M).

We observed that lysosomes were predominantly tubular during early aging, whereas enlarged vesicular lysosomes became more prevalent in late aging. We propose that the different cellular conditions during early and late aging stages

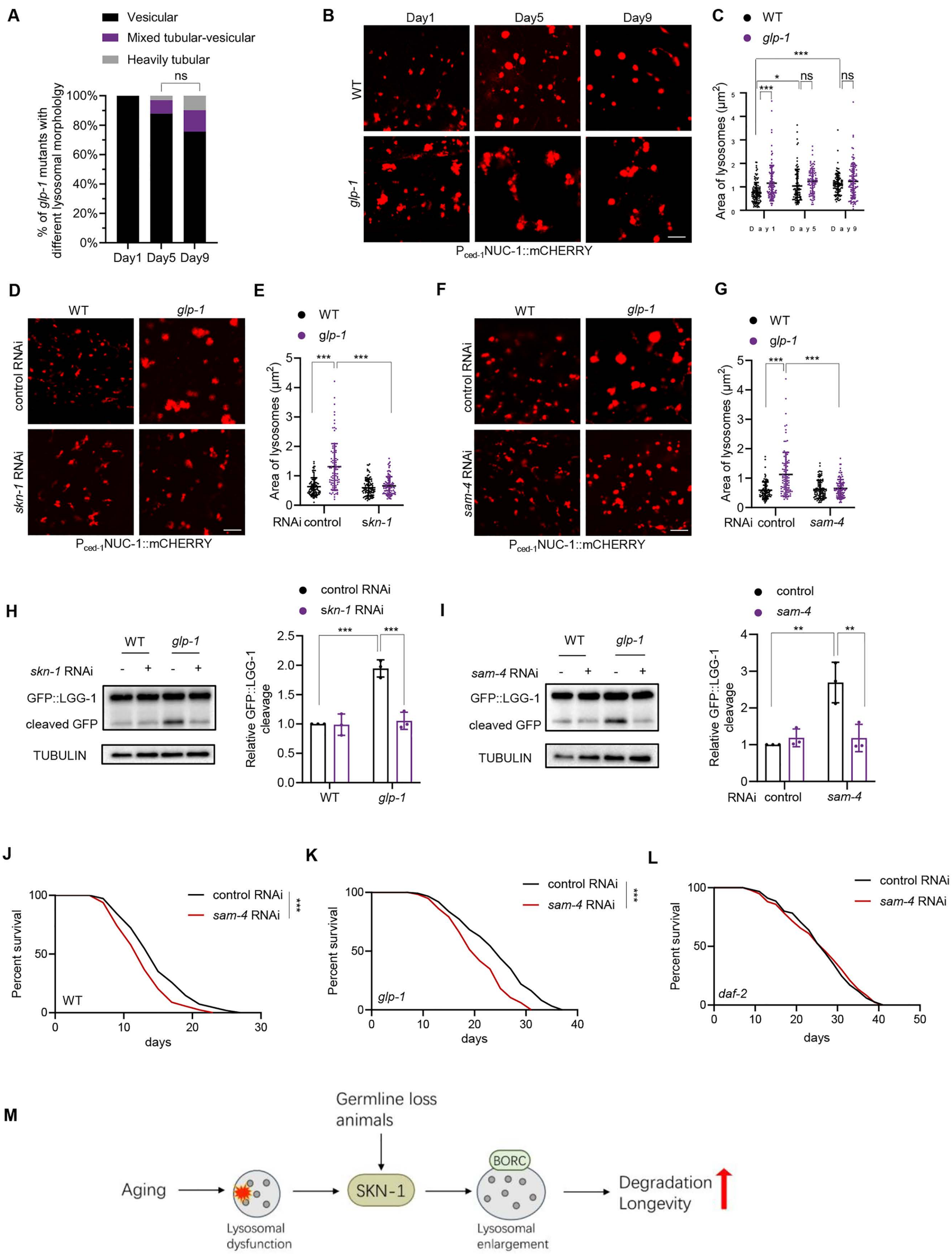
A
Vesicular
Mixed tubular-vesicular
Heavily tubular
ns
% of glp-1 mutants with different lysosomal morphology
Day1 Day5 Day9
B
Day1 Day5 Day9
WT
glp-1
Pced-1NUC-1::mCHERRY
C
WT
glp-1
Area of lysosomes (μm2)
Day1 Day5 Day9
D
WT glp-1
control RNAi
skn-1 RNAi
Pced-1NUC-1::mCHERRY
E
WT
glp-1
Area of lysosomes (μm2)
RNAi control skn-1
F
WT glp-1
control RNAi
sam-4 RNAi
Pced-1NUC-1::mCHERRY
G
WT
glp-1
Area of lysosomes (μm2)
RNAi control sam-4
H
WT glp-1
skn-1 RNAi
GFP::LGG-1
cleaved GFP
TUBULIN
control RNAi
skn-1 RNAi
Relative GFP::LGG-1 cleavage
WT glp-1
I
WT glp-1
sam-4 RNAi
GFP::LGG-1
cleaved GFP
TUBULIN
control
sam-4
Relative GFP::LGG-1 cleavage
RNAi control sam-4
J
control RNAi
sam-4 RNAi
Percent survival
WT
days
K
control RNAi
sam-4 RNAi
Percent survival
glp-1
days
L
control RNAi
sam-4 RNAi
ns
Percent survival
daf-2
days
M
Germline loss animals
Aging
Lysosomal dysfunction
SKN-1
BORC
Lysosomal enlargement
Degradation
Longevity

**Fig 6. SKN-1-mediated lysosomal enlargement is crucial for lysosomal function and longevity. (A)** Ratio of *glp-1* mutants exhibiting different lysosomal morphologies in the hypodermis during aging. Chi-square and Fisher's exact test. $n = 33-41$ animals. **(B and C)** Vesicular lysosomal morphology (B) and size (C) in *glp-1* mutants in the hypodermis during aging. Two-way ANOVA analysis followed by Tukey's multiple comparisons post hoc test. $n = 83-111$ lysosomes. **(D and E)** Effect of *skn-1* RNAi on lysosomal morphology (D) and size (E) in the hypodermis of day 1 *glp-1* mutants. Two-way ANOVA analysis followed by Tukey's multiple comparisons post hoc test. $n = 100$ lysosomes. **(F and G)** Effect of *sam-4* RNAi on lysosomal morphology (F) and size (G) in the hypodermis of day 1 *glp-1* mutants. Two-way ANOVA analysis followed by Tukey's multiple comparisons post hoc test. $n = 100$ lysosomes. **(H and I)** Effect of *skn-1* RNAi (H) or *sam-4* RNAi (I) on GFP::LGG-1 cleavage in day 1 *glp-1* mutants. Left: representative images. Right: quantification data. Two-way ANOVA analysis followed by Tukey's multiple comparisons post hoc test. $n =$ three independent experiments. **(J and L)** Effect of *sam-4* RNAi on the life span of WT (J), *glp-1* mutants (K), and *daf-2* mutants (L). Log-rank (Mantel–Cox) test s followed by Bonferroni post hoc tests. Statistical analysis and additional repeats were listed in S1 Table. **(M)** Working model of the regulation and function of lysosomal enlargement. Lysosomal enlargement occurs as a response to functional decline during aging. Upon lysosomal dysfunction, the transcription factor SKN-1 is activated to promote lysosomal enlargement by limiting fission in a BORC-dependent manner. This enlargement enhances lysosomal degradation capacity, serving as a beneficial adaptive response. In long-lived *glp-1* mutants, SKN-1 activation similarly drives lysosomal enlargement, boosting degradation capacity and contributing to their extended life span. Data are presented as mean ± SD. *$p < 0.05$, **$p < 0.01$, ***$p < 0.001$. Scale bar = 5 μm for panels (B), (D), and (F). The numerical data presented in this figure can be found in S1 Data. Immunoblot raw images in this figure can be found in S1 File.


determine the corresponding lysosomal morphology. In early aging, the gradual accumulation of protein aggregates and damaged organelles leads to elevated autophagic demand, which promotes lysosomal tubulation. Meanwhile, most lysosomes remain functional during this phase and do not require size enlargement. Hence, tubular lysosomes predominate in early aging. As aging progresses, however, an increasing number of lysosomes become dysfunctional. The lysosomal stress response is then prioritized to cope with their own functional impairments, resulting in an increase in lysosomal size. Supporting this model, basal SKN-1 activity, as measured by expression of its target *gst-4*, was increased in late aging compared to early aging [33], which correlates lysosomal enlargement with the activity of its central regulator SKN-1 in late aging. Therefore, the alterations of lysosomal morphology reflect a dynamic adaptation of lysosomal morphology to the changing cellular environment during aging.

The mechanism by which lysosomal enlargement improves their function remains unclear. Lysosomal degradation depends on the activity of degradative enzymes, which is impaired during lysosomal dysfunction. In addition to enzyme activity, the degradation capacity of individual lysosomes is also influenced by their enzyme content, which could be theoretically increased by lysosomal enlargement. It is possible that aged or dysfunctional lysosomes may compensate for impaired function by increasing their size, thereby accommodating more enzymes and potentially enhancing their degradation capacity.

Enlarged lysosomes may differ in their membrane composition from regular lysosomes, which might be the molecular basis for their morphological changes. We found that restricted fission may be the cause of lysosomal enlargement. Studies have shown that lysosomal fission is regulated by specific membrane lipids, such as phosphatidylinositol-3,5-bisphosphate (PtdIns(3,5)$P_2$) [24], PtdIns(4,5)$P_2$ [34], and C22 glucosylceramide [23]. Therefore, lysosomal enlargement may be accompanied by alterations in membrane lipid composition. Interestingly, both our laboratory and others have found that an important function of SKN-1 is to regulate lipid metabolism [31,35], leading to the hypothesis that SKN-1 promotes lysosomal enlargement by modulating lipid metabolism and thereby altering lysosomal membrane lipid composition.

The key regulator of lysosomal enlargement is the transcription factor SKN-1, which is best known for its role in maintaining cellular redox homeostasis [17]. This study establishes a novel link between SKN-1 and lysosomal homeostasis, demonstrating that SKN-1 not only responds to changes in lysosomal function but also actively maintains lysosomal function by regulating lysosomal morphology. These findings suggest that SKN-1 may sense lysosomal dysfunction, raising the possibility that lysosome-associated factors play a pivotal role in regulating SKN-1 activity. Therefore, further exploration of the molecular pathways upstream of SKN-1 in the context of lysosomal dysfunction will not only advance our understanding of lysosomal surveillance but may also uncover broader mechanisms that govern SKN-1 activity.

Enlarged lysosomes are often associated with various diseases. Prolonged inhibition of fission can disrupt lysosomal reformation and compromise the lysosomal pool, ultimately impairing lysosomal function [23,36]. This notion may appear

contradictory to our findings. We propose that prolonged enlargement resulting from restricted fission is detrimental, as it progressively reduces the lysosomal pool. In contrast, short-term lysosomal enlargement may serve as an adaptive stress response to cope with acute dysfunction. Interestingly, the long-lived mutant *glp-1* exhibit sustained lysosomal enlargement over time. We speculate the presence of a compensatory mechanism that preserves the lysosomal pool, potentially involving TFEB/HLH-30-mediated lysosomal biogenesis. Indeed, TFEB/HLH-30 is activated in *glp-1* mutants and is required for their longevity [18]. Therefore, we suggest that sustainable enhancement of lysosomal function through enlargement requires concomitant maintenance of lysosomal quantity.

## Materials and methods

### *C. elegans* strains and maintenance

*C. elegans* were cultured on standard nematode growth medium (NGM) seeded with *Escherichia coli (E. coli)* OP50−1 [37]. The following strains were provided by Caenorhabditis Genome Center: wild-type N2 Bristol, CB4037[*glp-1(e2141)*], CB1370[*daf-2(e1370)*], CL2166[*gst-4p*::*gfp*], DA2123[*lgg-1p::gfp::lgg-1*], MAH215[*lgg-1p::mCherry::gfp::lgg-1*], MGH171[*sid-1(qt9); alxIs(vha-6p::sid-1::SL2::GFP)*], and VC1772[*skn-1(ok2315)*]. SPC207[*skn-1(lax120)*] was provided by Dr. Sean Curran. *qxIS257*(*ced-1p::nuc-1::mcherry*) and *qxIS354(ced-1p::laat-1::gfp)* strains were provided by Dr. Xiaochen Wang. The strain expressing *sam-4::gfp* in the hypodermis was generated in the author's laboratory by cloning the *dpy-7* promoter region along with the full-length *sam-4* genomic sequence into pPD95.79 vector. The strain expressing *spin-1::mcherry* was generated in the author's laboratory by cloning the *spin-1* promoter region along with the full-length *spin-1* genomic sequence into pPD95.79 vector. The hypodermis-specific and muscle-specific RNAi strains were generated in the author's laboratory by crossing *rde-1(mkc36)* with *kzIs9(lin-26p::rde-1)* and *neIs9(myo-3::HA::rde-1)*, respectively. Double mutants were created using standard genetic techniques. The *glp-1(e2141)* mutants were raised at 25 °C during the larval stage to induce sterility and prolong life span.

### Microbe strains and RNAi treatment

*E. coli* OP50−1 bacteria were cultured overnight at 37 °C in LB and then seeded onto NGM plates. For RNAi experiments, HT115 bacteria containing specific dsRNA-expression plasmids (Ahringer library) [38] were cultured overnight at 37 °C in LB supplemented with 100 μg/ml carbenicillin and seeded onto NGM plates containing 5 mM IPTG [39]. RNAi was induced at room temperature for 24 hours prior to adding L1-stage worms to the RNAi plates to knock down the targeted genes. Phenotypes were then examined at the day 1 adult-stage unless otherwise specified in figure legends. The genes in the TF RNAi library were listed in S2 Table.

### Chemical treatments

TBHP was diluted to 75 mM with M9 buffer, and 100 μL of this solution was added to NGM plates to yield a final concentration of 1.75 mM. LLOME was diluted to 500 mM with DMSO, and 60 μL of this solution was added to NGM plates to yield a final concentration of 7.5 mM. $H_2O_2$ was diluted to 60 mM with M9 buffer, and 100 μL of this solution was added to NGM plates to yield a final concentration of 1.5 mM. NAC was diluted to 610 mM with M9 buffer, and 17 μL of this solution was added to NGM plates to yield a final concentration of 1 mM. Day 1 adult-stage worms were transferred to these plates and treated for durations specified in figure legends.

### Fluorescent microscopy

To analyze GFP or mCHERRY fluorescence, adult worms were paralyzed using levamisole or $NaN_3$, and mounted on slides. The mCHERRY::GFP::LGG-1 strain was specifically paralyzed using $NaN_3$, as levamisole was reported to enhance GFP::LGG-1 puncta formation [40]. Fluorescence images were acquired using Nikon NIS-Elements software or Leica LAS

X software. Lysosomal size was determined by using the "Quantify" tool in Leica LAS X software. To specifically measure vesicular lysosome size, only animals exhibiting predominantly vesicular lysosomes were analyzed. The "draw polygon" and "draw ellipse" functions in the Leica LAS X software were employed to manually circle vesicular lysosomes while manually excluding tubular lysosomes. ROI areas were exported for size analysis.

The GFP/mCHERRY ratio was calculated in ImageJ 1.53e by selecting the "Green" and "Red" channels to quantify respective fluorescence intensities. For *gst-4p::*GFP intensity measurements, images were processed in ImageJ 1.53e by selecting the green channel, delineating worm bodies using freehand selections, and exporting intensity values.

### Examination of lysosomal phenotypes

Lysosomal size was quantified by analyzing lysosomes in the midbody region of worms. Lysosomal number was counted within an area of 20 × 20 μm in the midbody region of worms. To examine lysosomal fission, time-lapse images were captured every 1 s for 1 min. Fission events were analyzed within an area of 50 × 50 μm per worm and were quantified by the percentage of worms exhibiting fission events within a one-minute observation period. To calculate GFP/mCHERRY ratio in the mCHERRY::GFP::LGG-1 strain, images from a 10 × 10 μm area per worm were analyzed. To assess lysosomal acid phosphatase activity, ~1,000 day-1 adult worms were collected, sonicated, and subjected to enzyme activity measurement using an Acid Phosphatase Assay Kit (P0326, beyotime), and enzyme activity was normalized to the protein content.

For lysosomal size measurements, the presented data are representative of at least three independent experiments with similar results, each comprising six worms per group. For lysosomal number, fission quantification, and GFP/mCHERRY ratios, the data represent pooled results from at least three independent experiments, totaling 19–35 worms per group. Acid phosphatase activity data represent pooled results from three independent experiments.

### Immunoblotting

Immunoblotting assays were performed as previously described [41]. Day-1 adult worms were harvested and sonicated in RIPA buffer (100 mM Tris, pH 8.0, 150 mM NaCl, 1% Triton X-100, 1% deoxycholic acid, 0.1% SDS, 5 mM EDTA, and 10 mM NaF) supplemented with 1 mM DTT and proteinase inhibitors (Beyotime). Subsequently, the samples were boiled and loaded onto the gel. Antibodies targeting GFP (Santa Cruz, SC-9996, 1:2000), mCHERRY (Sungene Biotech, KM8,017, 1:2000), and TUBULIN (Sigma, T9026, 1:4000) were employed for immunoblotting. The images were quantified using Image J 1.53e. The cleavage of GFP::LGG-1 and NUC-1::mCHERRY was quantified by calculating the ratio of cleaved GFP or mCHERRY to total GFP or mCHERRY, respectively.

### Lifespan analysis

Lifespan assays were performed as previously described [42]. Briefly, synchronized L1-stage worms were introduced to NGM plates supplemented with different *E. coli* strains. For the life span assay of *glp-1* mutants, temperatures were maintained at 25 °C from the L1 larval stage to day-1 adulthood, followed by a shift to 20 °C. Throughout the reproductive phase, worms were transferred daily. Instances where worms died of vulva burst, bagging, or crawling off the plates were censored from the analysis. Experimental replicates and statistical analysis are provided in S1 Table.

### Statistical analysis

Data are presented as mean ± SD. Statistical analysis was performed using GraphPad prism software. Student *t* test was performed when comparing two groups. One-way ANOVA was applied when multiple treatments belonged to a single factor. Two-way ANOVA was used when two independent factors coexisted. Chi-square and Fisher's exact test were employed for proportional data. Log-rank (Mantel–Cox) test was utilized for life span analysis. $p < 0.05$ was considered significant. Micrographic and immunoblotting images are representative of at least three independent experiments with similar results. The experimenters were not blinded.

## Supporting information

**S1 Fig. Aging is associated with lysosomal enlargement in the intestine and muscle.** (A) Ratio of animals exhibiting different lysosomal morphologies in the intestine during aging. Chi-square and Fisher's exact test. *n* = 28–34 animals. (B) The morphology and size of vesicular lysosomes in the intestine during aging. Left: representative images. Right: quantification data. One-way ANOVA analysis followed by Dunnett's multiple comparisons post hoc test. *n* = 84–88 lysosomes. (C) The morphology and size of vesicular lysosomes in the muscle during aging. Left: representative images. Right: quantification data. One-way ANOVA analysis followed by Dunnett's multiple comparisons post hoc test. *n* = 90–102 lysosomes. Data are presented as mean ± SD. *$p < 0.05$, ***$p < 0.001$. Scale bar = 5 μm for panels (B) and (C). The numerical data presented in this figure can be found in S1 Data.
(TIF)

**S2 Fig. Lysosomal enlargement in response to aging-related lysosomal dysfunction.** (A) Effect of chronic TBHP treatment on the activity of lysosomal acid phosphate in day 1 adults. One-way ANOVA analysis followed by Dunnett's multiple comparisons post hoc test. *n* = three independent experiments. (B and C) Effect of 6-hour hydrogen peroxide treatment on lysosomal morphology (B) and size (C) in the hypodermis of day 1 adults. Unpaired *t* test analysis. *n* = 100 lysosomes. (D–F) Effect of 6-hour TBHP treatment on the GFP/mCHERRY ratio in the hypodermis (D), intestine (E), and muscle (F) of mCHERRY::GFP::LGG-1 day 1 adults. Left: representative images. Right: quantification data. Unpaired *t* test analysis. *n* = 21 animals. (G–J) Effect of 6-hour TBHP treatment on lysosomal morphology and size in the intestine (G, H) and muscle (I, J) of day 1 adults. (G, I): representative images. (H, J): quantification data. Unpaired *t* test analysis. *n* = 83–91 lysosomes for (H) and 90 lysosomes for (J). (K) Effect of LLOME treatment on the activity of lysosomal acid phosphate in day 1 adults. One-way ANOVA analysis followed by Dunnett's multiple comparisons post hoc test. Data are presented as mean ± SD. **$p < 0.01$, ***$p < 0.001$. Scale bar = 5 μm for panels (B), (G), and (I); 1.25 μm for panels (D–F). The numerical data presented in this figure can be found in S1 Data.
(TIF)

**S3 Fig. SKN-1 regulates lysosomal enlargement.** (A and B) Effect of *skn-1(ok2315)* mutation on lysosomal morphology (A) and size (B) in response to TBHP treatment in the hypodermis of day 1 adults. Two-way ANOVA analysis followed by Tukey's multiple comparisons post hoc test. *n* = 93−99 lysosomes. (C and D) Effect of *hlh-30* RNAi on lysosomal morphology (C) and size (D) in response to 6-hour TBHP treatment in the hypodermis of day 1 adults. Two-way ANOVA analysis followed by Tukey's multiple comparisons post hoc test. *n* = 100−132 lysosomes. (E and F) Effect of *skn-1(ok2315)* mutation on vesicular lysosomal morphology (E) and size (F) in the hypodermis during aging. Two-way ANOVA analysis followed by Tukey's multiple comparisons post hoc test. *n* = 90−104 lysosomes. (G) Effect of *skn-1* RNAi on the ratio of animals exhibiting different lysosomal morphologies in the hypodermis during aging. Chi-square and Fisher's exact test. *n* = 54−67 animals. (H) 6-hour TBHP treatment induces expression of *gst-4p*::GFP in day 1 adults. Left: representative images. Right: quantification of fluorescent intensity. Unpaired *t* test analysis. *n* = 20 animals. (I) 6-hour LLOME treatment induces expression of *gst-4p*::GFP in day 1 adults. Left: representative images. Right: quantification of fluorescent intensity. Unpaired *t* test analysis. *n* = 27−28 animals. (J) Effect of *skn-1 gof* mutation on the ratio of animals exhibiting different lysosomal morphologies in the hypodermis during aging. Chi-square and Fisher's exact test. *n* = 51−52 animals. (K and L) Effect of *wdr-23* RNAi on lysosomal morphology (K) and size (L) in the hypodermis of day 1 adults. Unpaired *t* test analysis. *n* = 100 lysosomes. Data are presented as mean ± SD. ***$p < 0.001$. Scale bar = 5 μm for panels (A), (C), (E), and (K); 150 μm for panels (H) and (I). The numerical data presented in this figure can be found in S1 Data.
(TIF)

**S4 Fig. BORC mediates lysosomal enlargement in response to lysosomal dysfunction.** (A and B) Effect of *sam-4* RNAi on lysosomal morphology (A) and size (B) in response to *skn-1 gof* mutation in the hypodermis of day 1 adults. Two-way

ANOVA analysis followed by Tukey's multiple comparisons post hoc test. $n = 100$ lysosomes. (C) Effect of *sam-4* RNAi on the ratio of animals exhibiting different lysosomal morphologies in the hypodermis during aging. Chi-square and Fisher's exact test. $n = 55$–63 animals. (D–I) Effects of hypodermis-specific (D, E), intestine-specific (F, G), and muscle-specific (H, I) *sam-4* RNAi on hypodermal lysosomal morphology and size in response to 6-hour TBHP treatment in day 1 adults. (D, F, and H): representative images. (E, G, and I): quantification data. Two-way ANOVA analysis followed by Tukey's multiple comparisons post hoc test. $n = 91$–97 lysosomes for (E), 89–93 lysosomes for (G), and 84–93 lysosomes for (I). (J) Effect of 6-hour TBHP treatment on lysosomal association with SAM-4::GFP in the hypodermis of day 1 adults. Left: representative images. White arrows indicate SAM-4::GFP signals. Right: quantification of lysosomal association with SAM-4::GFP. Chi-square and Fisher's exact test. $n = 100$ lysosomes. (K) Effect of *skn-1 gof* mutation on lysosomal association with SAM-4::GFP in the hypodermis of day 1 adults. Left: representative images. White arrows indicate SAM-4::GFP signals. Right: quantification of lysosomal association with SAM-4::GFP. Chi-square and Fisher's exact test. $n = 100$ lysosomes. (L) Effect of *skn-1 gof* mutation on lysosomal fission in the hypodermis of day 1 adults. Chi-square and Fisher's exact test. $n = 19$–28 animals. (M and N) Effects of *skn-1* (M) and *sam-4* (N) RNAi on lysosomal number in response to 6-hour TBHP treatment in the hypodermis of day 1 adults. Two-way ANOVA analysis followed by Tukey's multiple comparisons post hoc test. $n = 26$–35 animals for (M), and 23–30 animals for (N). (O) Effect of *skn-1 gof* mutation on lysosomal number in the hypodermis of day 1 adults. Unpaired *t* test analysis. $n = 22$–26 animals. Data are presented as mean ± SD. ***$p < 0.001$. Scale bar = 5 μm for panels (A), (D), (F), and (H); 1.25 μm for panels (J) and (K). The numerical data presented in this figure can be found in S1 Data.
(TIF)

**S5 Fig. Autolysosomes are enlarged in the intestine and muscle in response to lysosomal dysfunction.** (A and B) Effects of *skn-1* and *sam-4* RNAi on AL size in the intestine (A) and muscle (B) of day 1 adults in response to 6-hour TBHP treatment. Left: representative images. Right: quantification data. Two-way ANOVA analysis followed by Tukey's multiple comparisons post hoc test. $n = 93$–98 lysosomes for (A) and 79–110 lysosomes for (B). (C and D) Effect of *skn-1* RNAi on AL size in the intestine (C) and muscle (D) during aging. Left: representative images. Right: quantification data. Two-way ANOVA analysis followed by Tukey's multiple comparisons post hoc test. $n = 89$–93 lysosomes for (C) and 92–106 lysosomes for (D). (E and F) Effect of *wdr-23* RNAi on AL size in the intestine (E) and muscle (F) of day 1 adults. Left: representative images. Right: quantification data. Unpaired *t* test analysis. $n = 90$–91 lysosomes for (E) and 89–95 lysosomes for (F). (G and H) Effect of *ppk-3* RNAi on AL size in the intestine (G) and muscle (H) of day 1 adults. Left: representative images. Right: quantification data. Unpaired *t* test analysis. $n = 85$–89 lysosomes for (G) and 83–93 lysosomes for (H). Data are presented as mean ± SD. **$p < 0.01$, ***$p < 0.001$. Scale bar = 2.5 μm for all panels. The numerical data presented in this figure can be found in S1 Data.
(TIF)

**S6 Fig. Lysosomal enlargement preserves lysosomal degradation capacity in the intestine and muscle.** (A and B) Effects of *skn-1* and *sam-4* RNAi on the GFP/mCHERRY ratio in intestine (A) and muscle (B) of mCHERRY::GFP::LGG-1 day 1 adults in response to 6-hour TBHP treatment. Left: representative images. Right: quantification data. Two-way ANOVA analysis followed by Tukey's multiple comparisons post hoc test. $n = 21$ animals. (C and D) Effect of *skn-1* RNAi on the GFP/mCHERRY ratio in the intestine (C) and muscle (D) during aging. Left: representative images. Right: quantification data. Two-way ANOVA analysis followed by Tukey's multiple comparisons post hoc test. $n = 21$ animals. (E-F) Effect of *wdr-23* RNAi on the GFP/mCHERRY ratio in the intestine (E) and muscle (F) of day 1 adults. Left: representative images. Right: quantification data. Unpaired *t* test analysis. $n = 21$ animals. (G and H) Effect of *ppk-3* RNAi on the GFP/mCHERRY ratio in the intestine (G) and muscle (H) of day 1 adults. Left: representative images. Right: quantification data. Unpaired *t* test analysis. $n = 21$ animals. Data are presented as mean ± SD. *$p < 0.05$, **$p < 0.01$, ***$p < 0.001$. Scale bar = 1.25 μm for all panels. The numerical data presented in this figure can be found in S1 Data.
(TIF)

**S7 Fig. The BORC subunit SAM-4 mediates life span in diverse tissues.** (A and B) Effect of *daf-2* mutation on lysosomal morphology (A) and size (B) in the hypodermis of day 1 adults. Unpaired *t* test analysis. $n = 100$ lysosomes. (C–E) Effects of hypodermis-specific RNAi (C), intestine-specific RNAi (D) and muscle-specific RNAi (E) of *sam-4* on the life span of WT animals. Log-rank (Mantel–Cox) test *s* followed by Bonferroni post hoc tests. Statistical analysis and additional repeats were listed in S1 Table. Data are presented as mean ± SD. $^*p < 0.05$, $^{**}p < 0.01$. Scale bar = 5 μm for panel (A). The numerical data presented in this figure can be found in S1 Data.
(TIF)

**S1 Table. Lifespan data.** Repeats 1 were graphed in figures.
(DOCX)

**S2 Table. List of transcription factors in RNAi screening.**
(XLSX)

**S1 Data. The underlying numerical data in the manuscript.**
(XLSX)

**S1 File. Original blots in this manuscript.**
(PDF)

## Acknowledgments

We acknowledge CGC, Dr. Sean Curran, and Dr. Xiaochen Wang for providing strains. We are grateful to the Analytic and Testing Center of Chongqing University for the use of the facility and technical support.

## Author contributions

**Conceptualization:** Haiqing Tang, Shanshan Pang.

**Data curation:** Xinyu Wang, Huimin Liu, Xiaoman Wang.

**Formal analysis:** Xinyu Wang, Huimin Liu, Xiaoman Wang.

**Funding acquisition:** Ben Zhou, Haiqing Tang, Shanshan Pang.

**Investigation:** Xinyu Wang, Huimin Liu, Xiaoman Wang.

**Methodology:** Xinyu Wang, Huimin Liu, Xiaoman Wang, Ben Zhou, Shanshan Pang.

**Project administration:** Haiqing Tang, Shanshan Pang.

**Resources:** Ben Zhou, Shanshan Pang.

**Supervision:** Ben Zhou, Haiqing Tang, Shanshan Pang.

**Writing – original draft:** Haiqing Tang, Shanshan Pang.

**Writing – review & editing:** Xinyu Wang, Huimin Liu, Haiqing Tang, Shanshan Pang.

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
