## [Editor Report · Decision Letter 0]

13 Jan 2025

Dear Dr Pang,

Thank you for submitting your manuscript entitled "SKN-1 Drives Lysosomal Enlargement to Maintain Function During Aging" for consideration as a Research Article by PLOS Biology.

Your manuscript has now been evaluated by the PLOS Biology editorial staff as well as by an academic editor with relevant expertise and I am writing to let you know that we would like to send your submission out for external peer review.

Once your full submission is complete, your paper will undergo a series of checks in preparation for peer review. After your manuscript has passed the checks it will be sent out for review. To provide the metadata for your submission, please Login to Editorial Manager (https://www.editorialmanager.com/pbiology) within two working days, i.e. by Jan 15 2025 11:59PM.

Kind regards,

Ines

--

Ines Alvarez-Garcia, PhD

Senior Editor

PLOS Biology

---

## [Decision Letter · Decision Letter 1]

1 Apr 2025

Dear Dr Pang,

Thank you for your patience while your manuscript entitled "SKN-1 Drives Lysosomal Enlargement to Maintain Function During Aging" was peer-reviewed at PLOS Biology and please accept my apologies for the delay in sending you our decision. The manuscript has now been evaluated by the PLOS Biology editors, an Academic Editor with relevant expertise, and by two independent reviewers.

The reviews are attached below. As you will see, the reviews find the conclusions interesting and novel however, they also raise several issues that would need to be addressed before we can consider the manuscript for publication. Both reviewers mention that the methods are unclear and miss important information that would allow to reproduce the work, thus you should expand and improve the methods significantly explaining the lysosomal quantification and analysing the potential function of tubular lysosomes and correlation with age. In addition, Reviewer 1 thinks you should quantify directly the effect on cargo turnover when enlarged lysosomes are clocked or enhanced, and also show that skn-1 KD decreases lysosomal activity in aging worms. Reviewer 2 mentions that the focus on a specific subset of lysosomes in the hypodermis is a limitation and that you should consider potential contributions of lysosomes from other tissues and quantify them to explore the function, and also asks if you can rule out the possibility that the effect on hypodermal lysosomes or lifespan is an indirect effect of SAM-4 activity in other tissues.

In light of the reviews, we would like to invite you to revise the work to thoroughly address the reviewers' reports. However, after discussions with the Academic Editor, we won't make Reviewer 1 point 7 essential for publication.

Given the extent of revision needed, we cannot make a decision about publication until we have seen the revised manuscript and your response to the reviewers' comments. Your revised manuscript is likely to be sent for further evaluation by all or a subset of the reviewers.

**IMPORTANT - SUBMITTING YOUR REVISION**

3. Resubmission Checklist

a) *PLOS Data Policy*

b) *Published Peer Review*

d) *Blurb*

Please also provide a blurb which (if accepted) will be included in our weekly and monthly Electronic Table of Contents, sent out to readers of PLOS Biology, and may be used to promote your article in social media. The blurb should be about 30-40 words long and is subject to editorial changes. It should, without exaggeration, entice people to read your manuscript. It should not be redundant with the title and should not contain acronyms or abbreviations. For examples, view our author guidelines: https://journals.plos.org/plosbiology/s/revising-your-manuscript#loc-blurb

Sincerely,

Ines

--

Ines Alvarez-Garcia, PhD

Senior Editor

PLOS Biology

Reviewers' comments

Rev. 1:

Lysosomes are crucial digestive organelles that preserve cellular homeostasis and functional decline of lysosomes has been associated with aging. Thus, understanding the mechanisms that might lead to their functional decline with age is a relevant topic. One age-dependent lysosomal change that has been documented in several recent studies is morphological changes, such as lysosome tubulation. In this study, Wang et al. describe another morphological change in aging lysosomes: enlargement of their vesicular morphology. The authors propose that this increase in lysosomal size results from lysosomal dysfunction and is a coping mechanism to prevent further lysosomal functional decline. They provide evidence that blocking the enlargement of dysfunctional lysosomes exacerbates their impairment further. Mechanistically, the authors show that the age-related enlargement of lysosomes is regulated by the longevity transcription factor, SKN-1, which limits lysosomal fission in response to lysosomal dysfunction to maintain lysosomal function. Overall, this paper presents interesting findings related to lysosome morphology changes that occur during the natural aging process and highlights the importance of lysosomal morphology adaptation to different stressors. However, there are some points that must be addressed to strengthen the quality of the work, rigor, and the relevance to the field.

Major points:

1. In general, there is not nearly enough description of the authors' methods used to obtain the data portrayed in the paper. As it is written, these results would not be able to be reproduced by any lab. First, they need to add more information about the various drugs used (concentration, solvent used for dilution, how much was spotted on each plate, when and for how long were the worms exposed). Second, they need to indicate the ages for all experiments in the figures and add more description about the tissues that were imaged in each experiment. Many of the panels describe the fluorescent marker, but not the promoter used. This information should be in the figure or legend. Lastly, they need more description of some of the analysis methods used. For example, in Figure 4I, they analyze the percentage of worms exhibiting fission, but do not describe whether they were looking at the whole worm or just a certain area. Additionally, throughout the paper, the number of lysosomes evaluated is reported but the number of worms where those numbers come from is not clear. This information is key for reproducibility and robustness of the findings.

2. Figure 1B shows that lysosomes also have a tubular morphology at day 5 and 9 of adulthood, but the authors do not address any of the tubular morphology in the rest of the paper. Additionally, the rest of the paper only shows vesicular lysosomes in experiments that were conducted at days 5 and 9, meaning these images were purposefully selected away from tubular lysosomes. I think it is misleading that the authors would selectively analyze only worms with vesicular lysosomes and ignore the other major morphological change that they describe in figure 1B. The authors need to examine and quantify tubular lysosomes throughout the rest of their lysosomal imaging experiments, but especially with skn-1 KD, skn-1 gof, and sam-4 KD. It would be informative to show if any of these conditions change proportion of vesicular and tubular lysosomes at various ages. These changes would give more specificity to how lysosomes change in age with skn-1 KD, skn-1 gof, and sam-4 KD.

3. On a related point, the authors only quantify vesicle size, but do not quantify vesicle number. In some of the representative images, it appears that as lysosomes enlarge, there may be fewer total vesicles (in skn-1 gof for example). Although it could be because they are morphing into tubular structures as well (it is hard to tell from some of the small images shown). The authors should quantify vesicle numbers to determine if there is a potential effect on lysosome biogenesis as well as quantify lysosome tubulation to appreciate the full impact on lysosomes.

4. The authors only measure lysosome functionality through GFP::LGG-1 cleavage and lysosomal acid phosphate activity. However, because these biochemical assays are done with tissue extracts, it does not inform much about the digestive capacity of the enlarged lysosomes specifically. The authors should directly quantify the effect on cargo turnover when enlarged lysosomes are blocked (skn-1 KD) or enhanced (skn-1 gof mutant). This could be done by using a tandemly tagged autophagy reporter strain (for example, mCherry::GFP::SQST-1) and quantifying GFP/mCherry ratio.

5. As an additional point, in the LGG-1 cleavage assays, the uncleaved and cleaved products should be shown in the same blot rather than cropping them out individually so that we can see the cleavage product relative to the uncleaved product in the same gel (similar to what has been done in other publications: PMID: 30929899).

6. The authors show that inducing early enlargement of vesicular lysosomes with ppk-3 RNAi increase activity and that skn-1 KD inhibits late-age lysosomal enlargement but never show that skn-1 KD reduces lysosomal activity in late-age worms. This is an essential experiment.

7. The authors show that ROS exposure (TBHP, hydrogen peroxide) and lysosomal membrane damage (LLOME) induce lysosomal enlargement. What about other cellular stresses, such as mitochondrial uncoupling (FCCP), ER stress (tunicamycin), and proteotoxic stress (MG132)? Would these other stressors induce skn-1 activation? These experiments would clarify whether dysfunction of other organelles induces a change in lysosome size.

8. This reviewer suggests validating the effect on lysosome enlargement with the skn-1 mutant strains available from CGC to control for possible skn-1 RNAi off-target effects.

9. In Figure 3A, the authors describe their screening strategy to identify transcription factor(s) that affect lysosomal enlargement, but this strategy is unclear. More description is needed about whether transcription factor RNAi was also used during the 6 hours that day 1 adults were exposed to TBHP. It would also be good to mention the total number of TFs that were screened.

10. The discussion could be significantly improved for clarity. For example, the last paragraph of the discussion section is very convoluted, and it is not clear what points are trying to be conveyed. The authors could also expand the discussion to discuss the levels of SKN-1 during aging and relate that to their data, speculate on the composition of enlarge lysosomes, and why lysosomes might enlarge instead of maintain the tubular network in late-age.

Minor points:

1. In the methods, the authors say they quantified gst-4p::GFP intensity using Image J software, but they need to add which features of the software they specifically used. They also need to include specific image settings that were used.

2. While describing lysosome fluorescent images in the results section, the authors need to indicate when those images were taken (age of adulthood). This also needs to be added to each figure caption. Similarly, in the methods section "Examination of lysosomal phenotype," the authors say they quantified lysosomal size using Leica LAS X software, but do not say which features they used in this software.

3. In line 57-58 of the introduction, the authors say, "lysosome enlargement is an organelle-autonomous response to their own dysfunction." They did not specifically prove this because they did not test whether other organelle/cellular stresses induced lysosome enlargement and/or skn-1 activation, so the wording of this sentence needs to be changed unless further experimentation is done.

4. This reviewer suggests migrating Figure S1B into main figure 1. Likewise, I would recommend moving the skn-1 gof data in Figure S4A-C into main figure 5 as this is an important observation that complements the skn-1 RNAi results.

5. The order of samples on the western blots sometimes does not match the order of the graphs (for example, 6H-I), which makes it confusing to interpret. I would recommend changing the order of the bar graphs to match the order of samples on the blots to make the presentation more straightforward.

Rev. 2:

The manuscript describes the enlargement of vesicular lysosomes during aging and in mutant long lived C. elegans nematodes. The authors use fluorescent microscopy techniques and RNAi to first characterize lysosomal enlargement during aging. They find that enlargement is an active response to lysosome disruption. This enlargement is mediated by snk-1 and if enlarging is inhibited lysosomal function is further impaired. Specifically long-lived glp-1 animals, but not other worms that have an extended lifespan, show enlarged lysosomes and this is important for the glp-1 mutation induced lifespan extension.

The data presented is interesting however it would be beneficial to consider the contribution of other tissues to the observed phenotypes. Additionally, the methods section lacks key details that are important for a clear understanding of the data by the reader.

1) One limitation of the study is the focus on a specific subset of lysosomes in the hypodermis. While this provides valuable insights, it would be beneficial to consider the potential contributions of lysosomes from other tissues. The study reports that a subset of hypodermal lysosome enlarges with age and the authors correlate size with lysosome function in whole worms. Several perturbations and measurements (RNAi, ROS exposures, western blots etc) are performed in whole worms and not in a tissue specific way. Thus, it is possible, and likely, that lysosomes in other tissues contribute to the observations.

For example, it seems that NUC-1::mCherry is expressed in other tissues as well (as per wormbase and single cell data). Thus, it is possible that the cleavage of the reporter is also caused by lysosomes other than the quantified hypodermal ones (Figure S1). Can the authors exclude the possibility that lysosomes from muscles or the intestine contribute to the signal? These might have a different response in terms of size. The same would apply to other interventions such as skn-1 RNAi or pkk-3 RNAi, which might not affect lysosome size in other tissues, but these lysosomes still contribute to the LGG-GFP cleavage signal (Figure 5). To be able to make the conclusion that enlargement is an active adaptive response to preserve lysosomal degradation capacity, lysosomes in other tissues should be quantified, tissue specific interventions should be used, or tissue specific reporters would be necessary.

To strengthen their conclusions, hypodermis specific image-based analysis could be performed by using the flux reporter mcherry::GFP::lgg (as in Figure 5) combined with lysotracker to stain for lysosomal acidity. This would allow to disentangle function at a cellular level and with excluding any potential impact of tubular lysosomes.

2) The methods section should be expanded to provide more details, ensuring that readers can fully understand the approach.

Lysosomal quantification: How is the area of lysosomes measured (manually, automatically, where in the worm e.g. head hypodermis, what is the depth imaged)? Are the tubular lysosomes excluded in the analysis and if so how (manually, were the investigators blinded)? How many worms are imaged per quantification? What happens to animals that show tubular structures (e.g. also in vha-7 RNAi) or animals that have both tubular and vesicular structures? Were those animals excluded from analysis? Are experiments merged and in case were there any batch effects? The authors use levamisole to fix the worms, which is known to cause LGG-1 puncta 5 minutes after mounting (PMID 25569839). At least some central experiments should be repeated using NaN3 immobilization to test if this shows similar phenotypes regarding their findings.

Tissue: The hypodermis is mentioned once in the context of Figure 1 and in context of the strain sam-4::GFP strain generation. It seems that the authors use the hypodermis for all quantifications. Maybe I missed this, but it would be helpful to have this information mentioned specifically in the methods but potentially also in the figure/legends.

Statistics: What is the criteria for showing a p-value versus not showing it? Could the authors please make it clear in the figure which conditions are compared separating the lines. Which pairwise comparison test is used within the ANOVA? Please elaborate on why the statistical tests switch between t-test, ANOVA and chi-square. Were experimenters blinded? Which test was used for the lifespan statistics (in the figure legend it states Mantel-Cox but in the table Bonferroni corrected is mentioned).

3) Some findings are based on sam-4 function as a myrlysine ortholog. The authors mention that further analysis is needed to confirm that sam-4 is located on lysosomal membranes. In addition to the missing GFP signal in the hypodermis, single cell RNAseq data shows only limited transcripts of sam-4 in the hypodermis and sam-4 RNAi has no effect on lysosomal size in unperturbed conditions. Can the authors rule out the possibility that the effect on hypodermal lysosomes or lifespan is an indirect consequence of SAM-4 activity in other tissues, such as impacting neuronal functions, or potentially other functions?

---

## [Decision Letter · Decision Letter 2]

15 Oct 2025

Dear Dr Pang,

Thank you for your patience while we considered your revised manuscript entitled "SKN-1 Drives Lysosomal Enlargement to Maintain Function During Aging" for publication as a Research Article at PLOS Biology. This revised version of your manuscript has been evaluated by the PLOS Biology editors, the Academic Editor and the two original reviewers.

Based on the reviews (pasted below), we are likely to accept this manuscript for publication, provided you satisfactorily address the remaining points raised by Reviewer 2. Please also make sure to address the data and other policy-related requests stated below my signature.

In addition, we would like you to consider a suggestion to improve the title:

"The transcription factor SKN-1 drives lysosomal enlargement during aging to maintain function"

We expect to receive your revised manuscript within two weeks.

*Published Peer Review History*

*Press*

Sincerely,

Ines

--

Ines Alvarez-Garcia, PhD

Senior Editor

PLOS Biology

DATA POLICY:

Many thanks for submitting the data underlying the graphs shown in the figures. I have checked the data and found several mistakes as follows - please correct them:

1) In Fig. 5, some values in the table seem to be mislabelled. As far as I can see, values for Fig. 5A are labelled as 5E; values for Fig. 5B as 5F; values for Fig. 5C as 5G values for Fig. 5E as 5A; values for Fig. 5F as 5B; values for Fig. 5G as 5C; Fig. 5D, H, I, J, K and L are all fine.

2) In S1 Data Fig. S7, please change the label of Fig. S7A data for B.

3) Please mention in Fig. S7 figure legend where the data can be found, as per the other figure legends.

CODE POLICY

We require the original, uncropped and minimally adjusted images supporting all blot and gel results reported in an article's figures or Supporting Information files. The ones you have sent us seem cropped, thus please carefully read our guidelines for how to prepare and resubmit this data: https://journals.plos.org/plosbiology/s/figures#loc-blot-and-gel-reporting-

requirements

SPECIES INDICATED IN THE ABSTRACT

- Please note that per journal policy, the model system/species studied should be clearly stated in the abstract of your manuscript.

Reviewers' comments

Rev. 1:

The authors have performed critical additional experiments that significantly improve the robustness of the work. Specifically, they have performed a much more rigorous analysis of all lysosome morphology and have examined autophagic turnover specifically at enlarged lysosomes and added essential details to their methodology. The addition of these critical experiments provides improved clarity on the functionality of the enlarged lysosomes. They have also significantly improved the discussion and presented a reasonable model. Overall, they have satisfactorily address all major concerns, improved the quality of the written manuscript, and I think it is now suitable for publication.

Rev. 2:

The authors have addressed my comments by including new experiments and additional information to the manuscript. I appreciate the authors' effort in including different tissues to test their contribution. The work showing that a subset of lysosomes enlarges with age as a compensatory mechanism is exciting, and I would recommend publication. However, in the revised version some minor comments have emerged which could be addressed with text edits.

(1) With the added information in the Methods section, it is now clear that data from multiple experiments were pooled. For some experiments (e.g. S5K, L and S6) it seems that each replicate has only few worms (4 individuals per replicate). This number seems low in comparison to other experiments especially since a small region (10x10µm) is analyzed. Could the authors please highlight the separate biological replicates in their data table (potentially also the graphs) to ensure that individual biological replicates are not skewing the results.

(2) The authors now clarified that they exclude the animals that show predominantly tubular structures from the image analysis. However, in the worm lysates western blots and the phosphatase assays, tubular lysosomes might contribute to the observations. Could the authors include a couple of sentences to acknowledge this limitation?

(3) The authors state in line 421 that a 100x100 µm area per worm was analyzed. Given that the worm is approximately 68µm wide this seems inconsistent. This might be a typo as later a 10x10 µm area is mentioned. Please clarify or correct this point.

(4) The acid phosphatase Kit P0326 that has now been described in the method section analyzes the activity of all acid phosphatases. What about acid phosphatases that are not in the lysosomes but e.g. secreted? The contribution of these might be low but it should be acknowledged. It should be considered to rephrase the statement made in the line 104 and 110-112 to highlight that other phosphatases outside of lysosomes might contribute to this result as well. In addition, it would be helpful to add how samples were normalized (by protein content?)

---

## [Editor Report · Decision Letter 3]

15 Nov 2025

Dear Dr Pang,

Thank you for the submission of your revised Research Article entitled "The Transcription Factor SKN-1 Drives Lysosomal Enlargement During Aging to Maintain Function" for publication in PLOS Biology. On behalf of my colleagues and the Academic Editor, William Mair, I am delighted to let you know that we can in principle accept your manuscript for publication, provided you address any remaining formatting and reporting issues. These will be detailed in an email you should receive within 2-3 business days from our colleagues in the journal operations team; no action is required from you until then. Please note that we will not be able to formally accept your manuscript and schedule it for publication until you have completed any requested changes.

PRESS

Sincerely, 

Ines

--

Ines Alvarez-Garcia, PhD

Senior Editor

PLOS Biology
